# New Geoeducational Facilities in Central Mazovia (Poland) Disseminate Knowledge about Local Geoheritage

**Maria Górska-Zabielska** 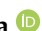

Institute of Geography and Environmental Sciences, Jan Kochanowski University, 25-406 Kielce, Poland; maria.gorska-zabielska@ujk.edu.pl

**Abstract:** Geoeducation is fundamental for safeguarding the abiotic world and its impact on the environment, which is inhabited by a society with ever-growing aspirations. However, current Earth and environmental science education in schools is insufficient. It requires creative and captivating methods that extend beyond traditional classroom settings, such as utilising new natural landscapes, in order to effectively implement geoeducation. New geological resources are unveiled during fieldwork or deep excavations. They can also be altered in situ through anthropogenic means to appear more visible to observers, particularly in remote tourism regions. As a geotourism product, these resources have the potential to serve as a catalyst for local economic growth. This article presents five geosites in central Mazovia, Poland, which were opened to the public in 2022 and 2023. Two Scandinavian erratic boulders, one of which has been developed, and three lapidaries with geotourism infrastructure are discussed. The research examines the significance of the erratic boulders for the natural and human environment.

**Keywords:** geoheritage; glacial boulder; lapidarium; geoeducation; geotourism; Masovian Lowland; central Poland

## 1. Introduction

The geological and geomorphological heritage of a region is known to researchers in the Earth and environmental sciences field who carry out methodically planned scientific research. Their results, published in professional journals and monographs (e.g., [1–6]) and/or at scientific conferences, rarely reach the inhabitants of the region or the tourists travelling there. Very often, people have no idea that there are sites of rich natural heritage in the immediate vicinity (e.g., [7]) that need to be preserved and developed.

Pantazopoulou et al. [8], based on the rich literature, state that the term "geoheritage" has evolved from the notion of "geological heritage". Heritage means something that has existed in the past, has been transmitted from the past and will continue to be important in the future. Geological heritage is understood as heritage that includes features of a geological nature. Global, regional and local geological elements such as igneous, metamorphic and sedimentary rocks, minerals, fossils, stratigraphic, tectonic, pedological, palaeontological structures and other geosites (or geotopes) at all scales are important objects/sites and specimens that provide information and insights into Earth formation, the development of life, past and present climates and landscapes, and the geological history of the places where they are found. The importance of geological monuments is equal to that of historical and archaeological monuments. Consequently, this type of information has scientific, educational, cultural, aesthetic and touristic value. It is therefore important to preserve it for present and future generations. The lack of this knowledge results in a deficit of geo-protection and a lack of care for such objects, which have important roles in the proper functioning of the environment. A serious issues remains for geography teachers who, with little sensitivity to the beauty of abiotic nature, fail to instil in their students a proper pro-environmental attitude. Despite the compulsory field classes (included in the

core curriculum; e.g., [9–11]), teachers, burnt out by didactic work, discuss geological and geomorphological issues ex catedra. Uninterested in their immediate surroundings, they do not know how to point out and discuss abiotic resources in simple language to those predisposed to absorbing knowledge at school age [12].

The qualitative research showed a complete lack of environmental awareness as well as of geo-ethical values [5,13,14]. Research on secondary school students has shown that students' education on geodiversity and geoheritage is significantly underdeveloped, so the integration of geoeducation into environmental education (geoenvironmental education) and the promotion of geological heritage with holistic approaches and interdisciplinary links are considered necessary [14,15]. The qualitative analysis showed that the implementation of an environmental programme may succeed in strengthening students' geocultural values. In addition, it appeared that students were empowered in issues and values of geocultural heritage and sustainability and developed feelings of environmental sensitivity.

The lack of geoeducation at school has serious consequences in the adult life of undereducated pupils. The lack of parental interest in exploratory family walks in the countryside, the widespread belief that geology is a difficult science and, unfortunately, the under-education of legislators mean that the citizen does not know what to protect and why. It is therefore not difficult to understand that they destroy geoheritage examples, cut them into slabs in a stonemason's workshop [16,17] or "decorate" them with graffiti. Beautiful and valuable geo-objects, witnesses of the ancient past of our small homelands, are disappearing irretrievably from our meadows and forests.

It seems, and this is also supported by the theses presented in the literature on the subject (e.g., [5,18,19]), that geoeducation is not as worth leaving school as is deepening it with elements of geo-protection. However, geoeducation must also be extended and offered in an attractive form to potential audiences outside of school. Only then will it help to understand how the environment works. Implementing guided interpretive tours, interactive displays, and educational workshops may help to deepen visitors' understanding of geoheritage and its delicate geoecosystem. Świeca [20] also suggests active forms of recreation into which elements of geoeducation can be woven. It can be included, for example, in the programmes of festivals for the whole family and in very popular museum nights. It can be implemented in urban summer nature walks [6], geocaching/quests or tourist recreational orienteering events such as TRInO [21]. Geoeducational elements are also present in ecomuseums, nature trails or thematic trails, not to mention geotouristic trails (e.g., [22–27]). Earth Day (21 April), Geodiversity Day (6 October), Geomorphologist Day (15 November), and even Geomorphology Week (1st week of March), which are already present in the Polish and international calendars, require much wider promotion on the part of the organisers and the geointerpreters involved, as information about these special initiatives currently reaches only a narrow circle of interested people.

An extremely important role in geoeducation is played by the geointerpreter or geotour guide [28]. He/she should remain faithful to the principle, given by Tilden (1957, cf. [29]), that "The guiding principle of effective interpretation is 'through interpretation, understand; through understanding, appreciate; through appreciation, protect'". Those responsible for geoeducation need to have a solid background in geology/geomorphology in order to understand the geological features and significance of the geosites they are interpreting to the public. A background in environmental education is essential to assess the ability of geosites to effectively communicate geological concepts to visitors. In addition to the obvious expertise, they must be able to reach an audience [7]. Among it, there will rarely be professionals, experts, professionally involved in the geosciences. Much more likely, there will be people with a passion for exploration tourism, but without specialist training. It is inevitable that they will also be people who walk along a nature trail without any special motivation, so to speak, and learn about inanimate nature issues along the way [30]. A special group are children who, on the one hand, have little knowledge or the typical enthusiasm and expertise of a child in relation to, for example, dinosaurs, and, on the other hand, are interested in their environment and have a strong imagination. With regard to

this group, developed leisure and tourism facilities, interactive exhibitions, games and activities related to the history of the earth are an excellent opportunity to achieve the educational goals of geotourism [12]. For each of these groups, a geo-interpreter wishing to achieve their educational objectives must be able to communicate their knowledge in a way that is appropriate to the age and involvement of the audience. His/her communication of geoheritage depends on the nature and characteristics of the target audience and their flexibility [29]. The more innovative and attractive the transfer, the more the guide affects the audience with his/her passion, enthusiasm, sometimes humour, and certainly references to physical processes in everyday life that are familiar to everyone (following the principle of "going from the familiar to the unknown" [28]), to mention the simplest ones such as cooking and baking, eating hot soup, cleaning with a vacuum cleaner), and the more effective the perception of the information will be.

The social responsibility of science through its innovative popularisation is a very current trend in today's world. Attractive geo-objects, by communicating specialised knowledge in an understandable but not rigorous or childish way (e.g., [7,31]), reveal the backstage of the ancient past, without destroying it, and preserving it for future generations. A sustainable approach to geoheritage is consistent with the principles of ecological security and contributes to improving the quality of life of all those who, in various ways, disseminate the resources and values of inanimate nature.

Geotourism is an effective driving force for the sustainable development of local government units located mainly in peripheral tourist areas. This type of tourism, which is not yet widespread in Poland, deals with providing tourist access to geological and geomorphological objects combined with geoeducation adapted to the age of the recipients and compulsory geoprotection (e.g., [6,32–41]). Geotourism brings economic benefits to local residents [42]. Entrepreneurs seeking to provide access to geological facilities should strive to utilise local labour, services, products, and supplies. Effective communication of the advantages of geotourism to the community encourages responsible destination management. Geotourism is deeply involved in local sustainable development.

## 2. Aim and Research Methods

The policy of acquiring new collections aimed at preserving the current state of inanimate nature [43] coincides with the authoress' research. The latest results of her investigations in central Mazovia allow her to present in this article five new geo-objects of glacial origin that are not known to the general public. They were exposed in bedrock sediments and still remain in situ or have been used for the construction of small facilities, which sometimes is called [44] an ex situ museum display of geology.

All the sites have great potential for geoeducation, for spreading knowledge about local geodiversity and for attracting the interest of geotourists and the general public. The authoress hopes that the geo-objects showcased among residents and tourists will benefit sustainable development in peripheral tourist locations, particularly the suburbs of major cities. The resonance and functionality of these objects are key factors in achieving this goal.

The fact that the suburbs of major Polish cities, including the capital, Warsaw, are not attractive for tourism in the usual sense of the word [45] does not mean that nature and educational tourism cannot develop there. In the light of research (e.g., [46–50]), residents often aspire to live in smart cities, characterised by innovative solutions and opportunities, where tourism, wellbeing and geoecosystem services meet the increasingly sophisticated needs of society. Finally, the prudent conservation of natural resources, facilitated by citizen involvement and expert assistance, has the potential to foster innovation, which is essential for sustainable economic growth and a high standard of living. The study provides an example of how five geo-objects can potentially address the incompatibility between universities, the dynamic market for ecosystem services, and the needs of local residents in the Mazovian region [51]. The approach shows promise for initial effectiveness.

The new geofacilities presented are erratic boulders. This type of geodiversity has been known in Mazovia for a long time, e.g., boulder "Mazur" from Piaseczno [52,53], or

an unnamed boulder from Długa Kościelna near Halinów [54]. Both of these exemplary exposures are protected by law as inanimate natural monuments, but unfortunately GDOŚ (Polish abbr. for General Directorate of Environmental Protection) does not include them in its register. From time to time, new Scandinavian boulders are discovered during construction works that penetrate the subsoil sediments to ever greater depths (e.g., in Warsaw in 2017 [55]).

The authoress' focus on two sites was initiated by local residents who wanted to highlight the geoheritage of their immediate surroundings. The following were single erratic boulders: an unnamed one in Żochy, Ojrzeń Commune, and Głaz "Jędrek" in Wilkowyja, Garwolin Commune. The remaining three geo-objects were presented by the authoress to local authorities and the local community as part of a public participation initiative. They are multifaceted sites in the form of a lapidarium in Pruszków, in Reguły and Komorów in Michałowice Commune.

The fieldwork was carried out using tried and tested survey methods (e.g., [6,32,35]). They consisted of the identification of erratic rocks, their petrographic type, and erratic type (based on own experience and available illustrated catalogues, e.g., [56–61]), and the measurement of dimensions. In the case of the idea to create a rock collection, geowatching [62] was carried out to select the available resources in order to find sufficiently large rock blocks with interesting microforms recorded on their surface, indicating the previous climatic conditions and geological processes that interacted with the erratic boulder from the parent area through glacial transport to the depositional environment. This study highlights the importance of analysing erratic boulders to better understand the history and evolution of glacial landscapes. Then, good models (e.g., [27]) were used at the stage of placing the geo-object at the target site. Each time, the objects were photographically archived according to the principles of [63].

In Poland, it is generally accepted that an erratic boulder is a rock fragment from the Scandinavian Shield and the Baltic Sea floor (Figure 1), transported and deposited by the Scandinavian ice sheet during one of the Pleistocene glaciations. In the literature [64–66], different dimensions of the erratics are given, according to which the fragment would be classified as a boulder. The variation in the length of the shortest axis of such an object is large, ranging from 0.256 m to just under 1 m according to the aforementioned authors.

Erratic boulders fulfil several functions, as recently pointed out (e.g., [6,32–37]). In order not to repeat the available information, but for the sake of order, only the functions, roles, significance and value of the erratic boulder will be briefly mentioned below. Its main aspect is its scientific value. No less important is its educational value and function, especially in the vicinity of schools. The skilful planning of the tourist development of such a geo-object, which is characterised by beauty and harmony with the surrounding environment, makes it an aesthetic value that is appreciated by recreationalists (recreational function) and of great importance in local sustainable development policy. The erratic boulders serve pro-environmental, cultural, and conservation purposes. This is something that has been very scarcely recognised by society in recent times.

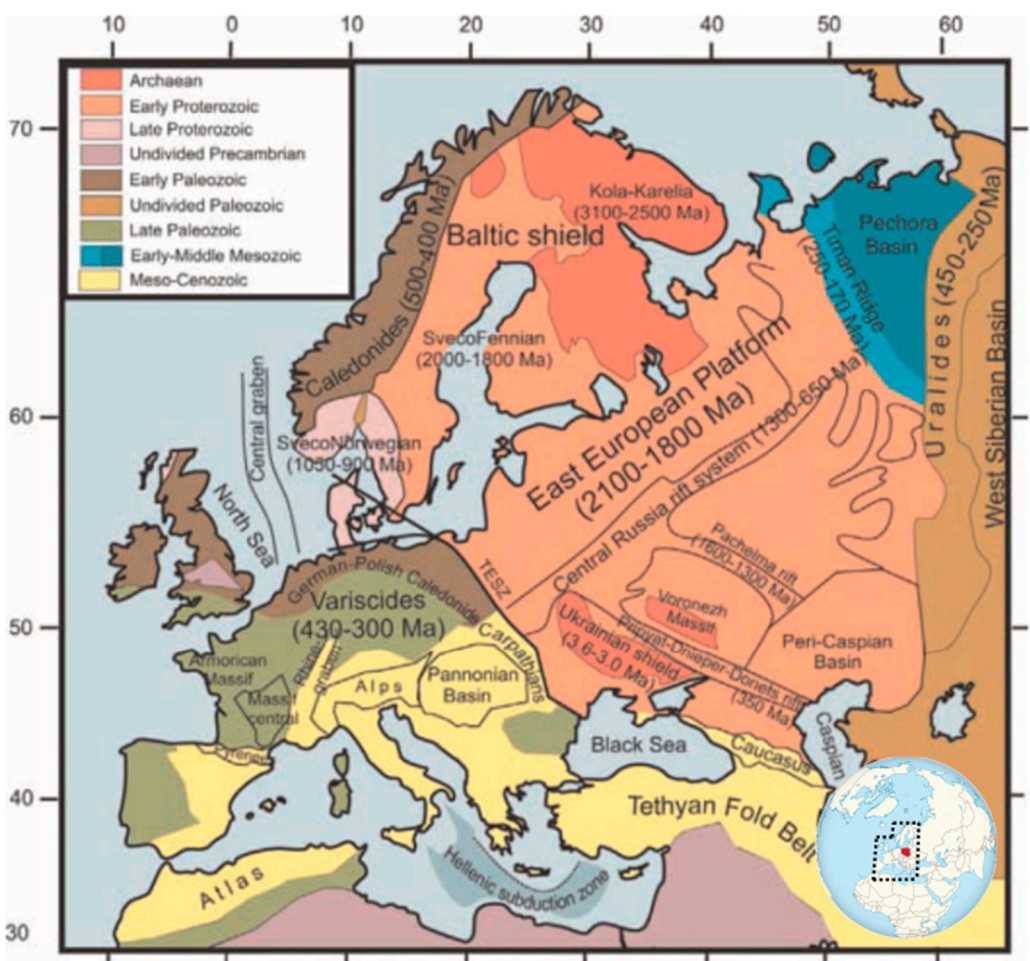

**Figure 1.** Schematic tectonic map of Europe with the location of Poland and an enlarged area shown in the bottom right corner. After [67], modified.

To protect them from damage and/or theft, erratic boulders are removed from their in situ position in urban controlled zones such as pocket gardens (e.g., [68]) or lapidaries/rock gardens (e.g., [6,24,34,36,69–73]).

### 3. Study Area

The geosites are located in the central part of Poland (Figure 2). Number 1 is situated within the Ciechanów Plain (no. 318.64 in the division of Poland into physical–geographical regions [74,75] in the North Masovian Lowland (316.6, ibidem)). The other geosites are located in the Middle Masovian Lowland (no. 318.7, ibidem): geosites 2–4 belong within the Łowicz-Błonie Plain (no. 318.72, ibidem) and the geosite 5 belongs to the Garwolin Plain (318.79, ibidem).

The relief of the study area was glacially formed during the Wartanian glaciation of the Central Polish Complex (MIS6 = Marine Oxygen-Isotope Stage [76,77]; Figure 2) and periglacially denuded when the Scandinavian Ice Sheet was found during its LGM (MIS2 [78,79]).

According to [75], the macro-region of the Central Mazovian Lowlands is characterised mainly by periglacial and fluvioglacial plains and undulating landscapes, with hills in places. These are interspersed with valley landscapes with dune hills, such as in the Vistula Valley. Pruszków and Michałowice are situated on denudational, periglacial and aeolian plains. Wilkowyja is a village located on an old glacial denudation plain.

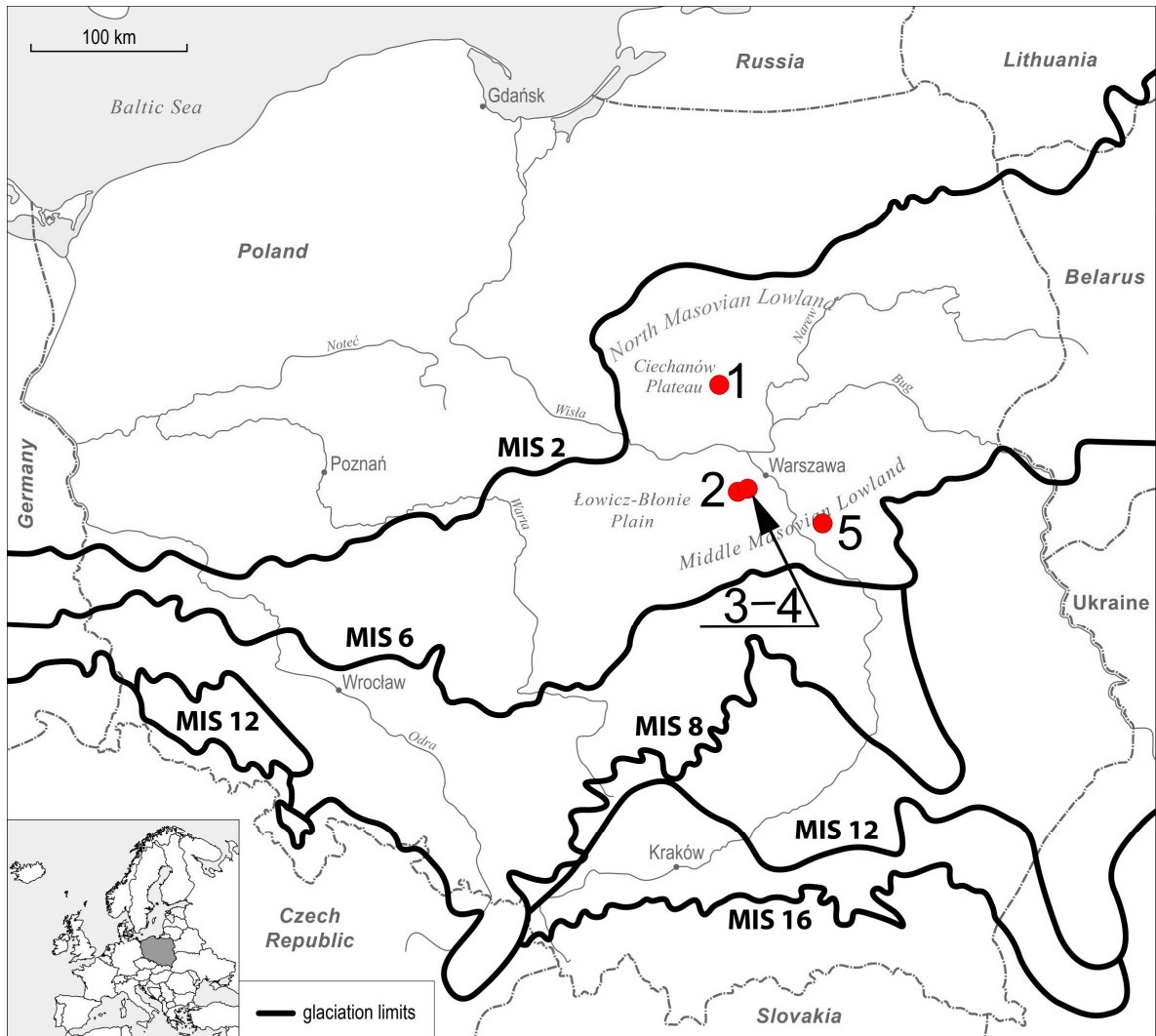

**Figure 2.** Location of geoeducational facilities of the North Masovian and Central Masovian Lowlands: 1—Żochy; 2—Pruszków; 3—Reguły; 4—Komorów; 5—Wilkowyja. Explanation of the abbreviations MIS 2—MIS 16: Marine Oxygen-Isotope Stages [78,79].

The predominant genetic relief types of the Ciechanów Upland (within the North Mazovian Lowland) are periglacial, denuded flat and undulating moraine uplands [75]. In its southern part, where Żochy is located, there are outliers of frontal moraines and kames from the recession of the Wkra stage of the Wartanian glaciation. The plateau is cut by river valleys with floodplains and overflow terraces. Dunes have developed on the latter.

All the erratic boulders in the study area were transported by the Scandinavian ice sheet during the same Wartanian glaciation, which occurred at about 180–120 ka BP [76,77]. The boulders may have been deposited during the advance of the ice sheet as it expanded and moved its front southwards, and they may have been abandoned during its retreat/contraction/melting as the front moved away to the north of the country. Therefore, it is difficult to determine the exact date of their deposition in different parts of Mazovia. Having emerged from under the melting ice mass, with abrasive edges typical of high-energy transport environments in glacial tunnels, the boulder load was subjected to the influence of periglacial processes in the foreland of the melting ice sheet. Traces of these processes, and of chronologically subsequent—contemporary—morphogenetic processes, are visible on the surface of all the geo-objects described in this article.

Five new geoeducational objects are located in three towns and two villages. They are single erratic boulders: unnamed (No. 1 in Figure 2) in Żochy, Commune Ojrzeń and

"Jędrek" (No. 5) in Wilkowyja, and Commune Garwolin. Multiple geofacilities in the form of a lapidarium can be found in Pruszków (No. 2) and in Reguły (No. 3) and Komorów (No. 4), in Commune Michałowice. All geo-objects are open to the public, although not all are easily accessible. Photographs were taken by the authoress, unless otherwise stated.

## 4. Overview of Geoeducational Facilities in Central Mazovia

### 4.1. An Unnamed Erratic Boulder in Żochy, Commune Ojrzeń, Ciechanów Poviat (No. 1 in Figure 2)

On the private arable land of a resident of the village of Żochy (52°46′20.2″ N; 20°34′41.6″ E), there is a huge erratic boulder in situ—granitic gneiss (Figure 3)—not completely preserved, but broken into five large parts (Figure 4). The upper part of one of them, which, according to the owner, protrudes one metre above the surface of the field, has always interfered with ploughing. Today, the boulder, which is largely exposed, shows its larger-than-average dimensions, the aesthetic value of which is unfortunately diminished by its broken silhouette.

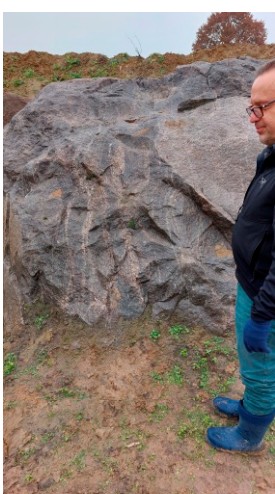

**Figure 3.** Details of geological structure of granitic gneiss in Żochy.

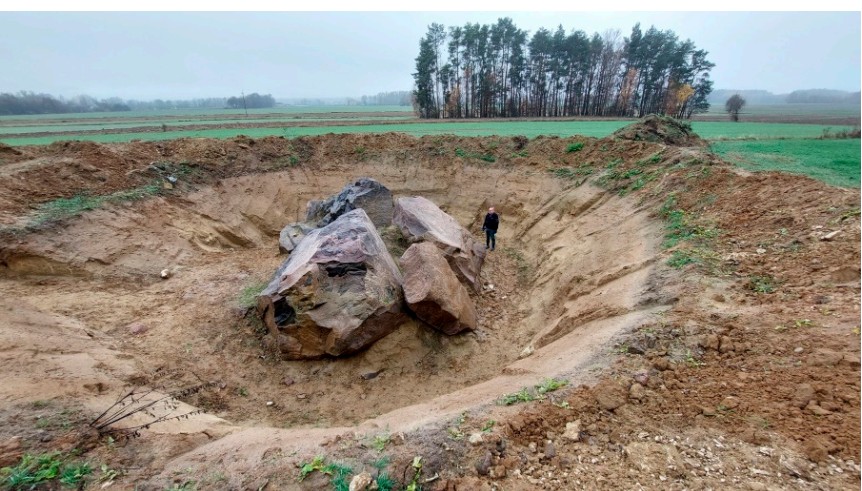

**Figure 4.** General view of fractured, largest erratic boulder of the Mazovian Voivodeship in Żochy.

The sum of the measured and calculated dimensions of all parts of the boulder gives the final values that characterise the granite gneiss. Its perimeter, 19.65 m, is the sum of the lengths of the outer walls of all parts, as if it were a single erratic boulder. Its volume is 56.27 m$^3$ and its weight is 154.75 tonnes. Comparing these dimensions with the dimensions

of the erratic boulders available in the GDOŚ list (as of December 2022), it turns out that this unnamed boulder from Żochy is the largest erratic boulder in the Masovian voivodship. For the sake of clarity, it is worth mentioning the dimensions of the Polish largest "Trigław" boulder, located in Tychów in Central Pomerania: a circumference of 43 m, a volume of 912 m$^3$ and a weight of 2508.6 tonnes.

The erratic boulder shows no traces of processes that took place in the parent area in Scandinavia, during glacial transport or in the periglacial zone on the foreland of the melting ice sheet. The crack is most likely of post-depositional origin.

Today, the erratic boulder is completely undeveloped and the owner has no idea how to present it to local people and possibly tourists. However, the fact that he initiated a conversation with the authoress about the future of the boulder shows that he is aware of the importance of such an object.

### 4.2. An Urban Inanimate Nature Trail in Pruszków (No. 2)

The inanimate nature trail (Figure 5) was created as part of the Pruszków City Budget 2022, a process of social consultation aimed at involving residents (including the authoress of this article) in deciding on a portion of the city's civic funding. The project aims to promote geotourism in Pruszków by establishing a facility with considerable educational value and recreational tourist infrastructure. The facilitator intends to enhance the walking experience as it will provide an outstanding platform to teach geology, geography, environmental protection, and sightseeing. In addition, it will impart positive attitudes towards pro-environmental and ecological values among residents and tourists while broadening their geological knowledge of the region. The inanimate nature trail has been created in accordance with the principles of the creation and management of footpaths (e.g., [27,80–82]). It is accessible to people with disabilities.

## PLAN OF INANIMATE NATURE PATH IN PRUSZKÓW

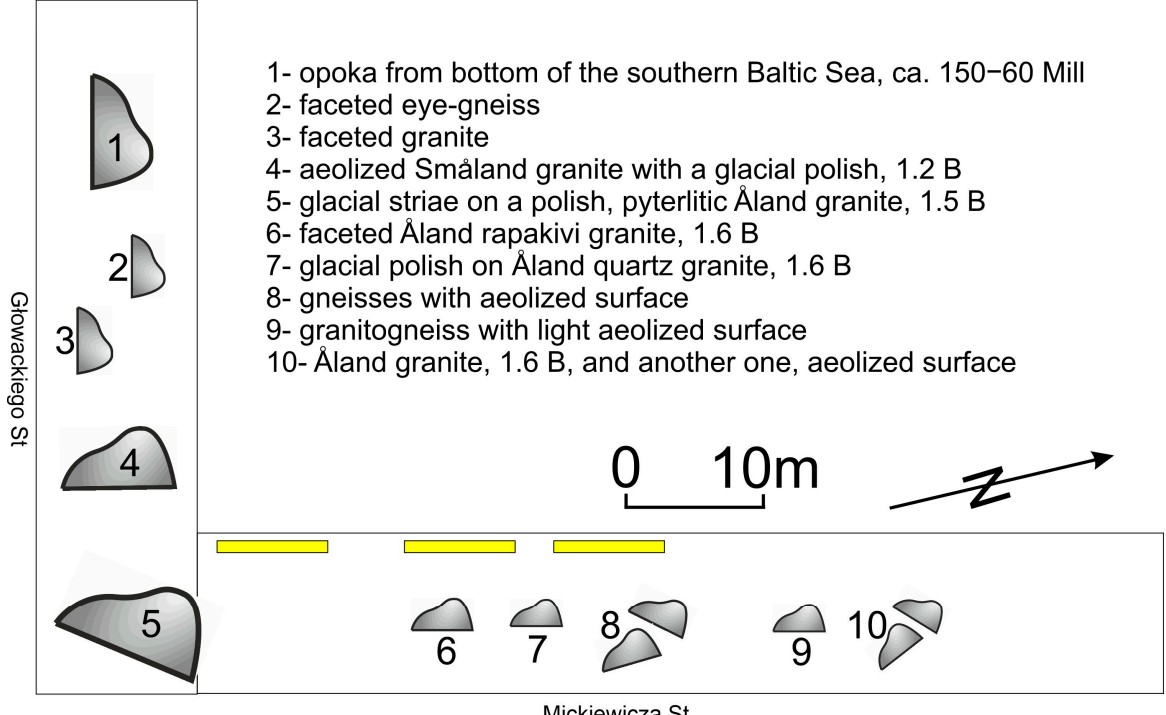

1- opoka from bottom of the southern Baltic Sea, ca. 150−60 Mill
2- faceted eye-gneiss
3- faceted granite
4- aeolized Småland granite with a glacial polish, 1.2 B
5- glacial striae on a polish, pyterlitic Åland granite, 1.5 B
6- faceted Åland rapakivi granite, 1.6 B
7- glacial polish on Åland quartz granite, 1.6 B
8- gneisses with aeolized surface
9- granitogneiss with light aeolized surface
10- Åland granite, 1.6 B, and another one, aeolized surface

**Figure 5.** Plan of the inanimate nature path in Pruszków. Yellow narrow rectangles stand in place of information panels.

There are twelve Scandinavian erratic boulders in the ten following sections, along the abiotic path in Pruszków (Figures 6 and 7), the map of which is shown in Figure 5. They were collected from within the city prior to being placed in their current location. In each case, permission was obtained from the existing owner/donor of the boulder to include it in the planned collection. Detailed information on the geological objects is given in Table 1 and their Scandinavian parent areas are given in Figure 8. The detailed characteristics of the erratic rocks along the nature trail are described by the authoress in a recent article [6]. There are igneous boulders (boulders No. 3–7, 10 in Figure 5) and metamorphic rocks (No. 2 and 8–9) from the Baltic Shield present in the collection. The collection also comprises a sedimentary rock (No. 1) from the layer covering the Baltic Shield. Both geological layers constitute the East European Platform in north-eastern Europe (Figure 1).

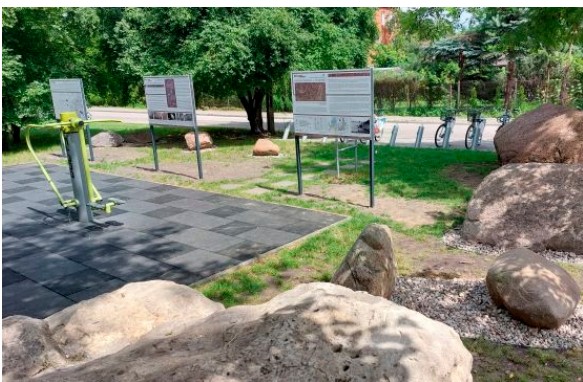

**Figure 6.** A nature trail located near the city bike stop and outdoor gym in Pruszków.

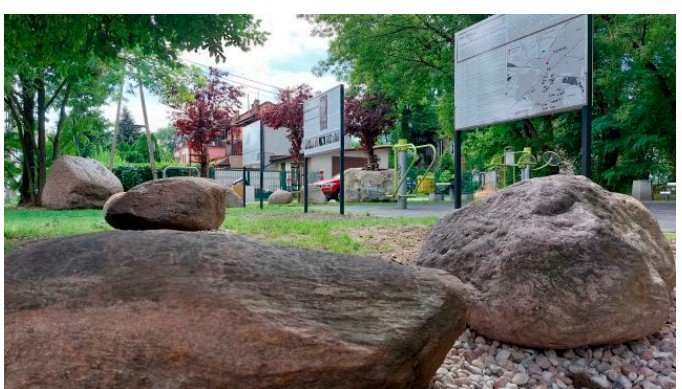

**Figure 7.** The nature trail consisting of a collection of 12 erratic boulders and three double-sided information boards.

**Table 1.** Information on erratic boulders on the inanimate nature trail in Pruszków.

| No. | Volume [m³] | Weight [t] | Petrographic Type of the Rock, Name of the Indicator Erratic and Its Scandinavian Provenance, Characteristics of the Specimen's Surface, Age |
|---|---|---|---|
| 1. | 1.59 | 4.36 | Opoka (sedimentary rock), bottom of the southern Baltic Sea; age: ca. 150–60 Mill |
| 2. | 0.16 | 0.43 | Eye-gneiss from the Baltic Shield, eolised surface relief, faceted rock; age: 1.2–1.5 B |
| 3. | 0.16 | 0.44 | Granite from the Baltic Shield, eolised surface relief, faceted rock; age: 1.2 B |
| 4. | 1.04 | 2.85 | Småland Granite (indicator erratic from SE Sweden), glacial polish, eolisation on the surface, lichen colonization; age: 1.2 B |
| 5. | 2.40 | 6.60 | Pyterlitic granite (indicator erratic from the Åland Islands), glacial striae on a polish; age: 1.5 B |
| 6. | 0.07 | 0.19 | Åland rapakivi granite (indicator erratic from the Åland Islands), faceted rock; age: 1.6 B |

Table 1. *Cont.*

| No. | Volume [m³] | Weight [t] | Petrographic Type of the Rock, Name of the Indicator Erratic and Its Scandinavian Provenance, Characteristics of the Specimen's Surface, Age |
|---|---|---|---|
| 7. | 0.16 | 0.44 | Åland quartz granite (indicator erratic from the Åland Islands), glacial polish; age: 1.6 B |
| 8. | 0.11 and 0.1 | 0.31 and 0.26 | Gneisses from the Baltic Shield, eolised surface relief, faceted rock; age: 1.2–1.5 B |
| 9. | 0.08 | 0.22 | Granito-gneiss from the Baltic Shield, light eolised surface relief; age: 1.2–1.5 B |
| 10. | 0.05 and 0.09 | 0.15 and 0.09 | Åland granite (indicator erratic from the Åland Islands), glacial polish, lichen colonisation; age: 1.6 B and another granite from the Baltic Shield |

Inventory: Maria Górska-Zabielska 2022.

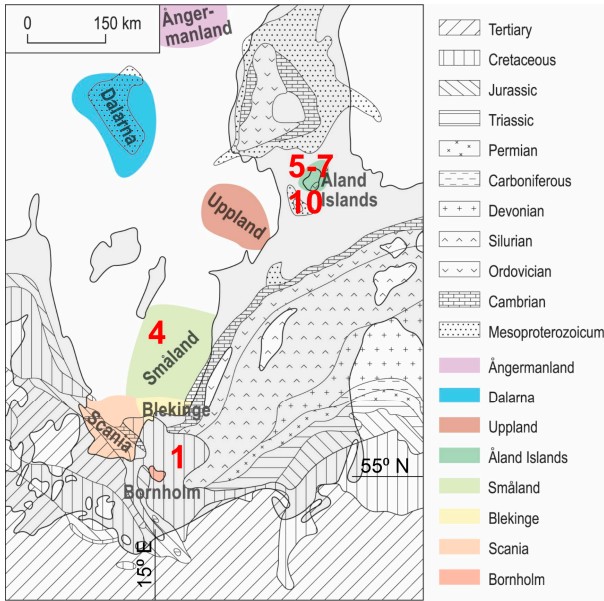

**Figure 8.** Indicator erratics' parent locations are situated on Pruszków's nature trail, depicted in an abstract manner against other primary Scandinavian feeding areas. Relevant numbers are listed in Table 1.

Erratic boulders currently on display along the nature trail have been encircled with small pebbles from Mazovian rivers purely for aesthetic purposes. Figure 9 offers a visual representation of the aforementioned formation. These pebbles are contemporaneous with the boulders since they also originated during the Wartanian glaciation, at approximately 130 ka BP, which is integral for the reader to understand the complete and coherent geological history of south-western Mazovia.

The erratic boulders exhibit is accompanied by three multicoloured, dual-faced panels expounding on separate aspects. Because they are aimed at local people, they are not bilingual. The panels have comparable layouts and a uniform editorial level. They relay the following topics: Written History within Stone (Figure 10), Significance of Erratic Rocks in Pruszków (Figure 11), and Erratic Rocks in the City (Figure 12). Each incorporates a QR code that redirects readers to more extensive information that could not fit on the panel's limited space. After reading the information on the display boards, visitors will understand the definition of a rock and its constituent minerals, as well as the origin of the erratic boulder found in Pruszków. One of the boards features a map of Pruszków, pinpointing the locations of other remaining erratic boulders within the town. Adjacent to the map is a table outlining the primary characteristics of these geological formations. The most fascinating specimens are depicted in the accompanying photographs. Furthermore, the components

of the rock carving which document geological and geomorphological processes and hold significance in terms of education are distinctly highlighted in the photographs (Figure 11).

The erratic boulder path is accompanied by additional tourist infrastructure in the form of city bike stations and an open-air gym (Figures 6, 7 and 9), where the inhabitants of Pruszków can practice active forms of recreation [20]. It is accessible to people with disabilities.

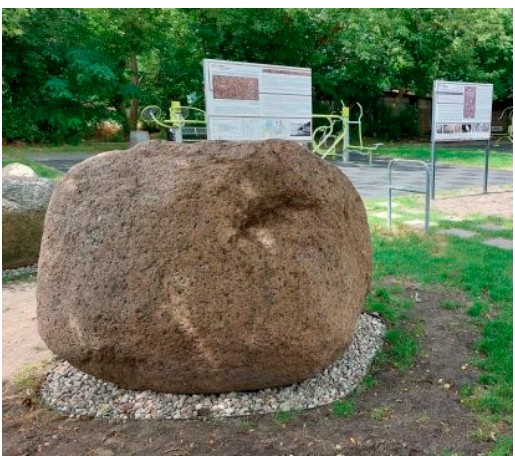

**Figure 9.** Mazovian river pebbles encircling a collection of erratic boulders.

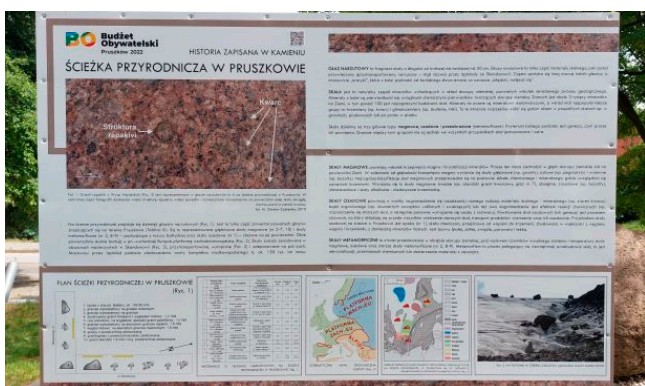

**Figure 10.** Information board entitled "Written History within Stone".

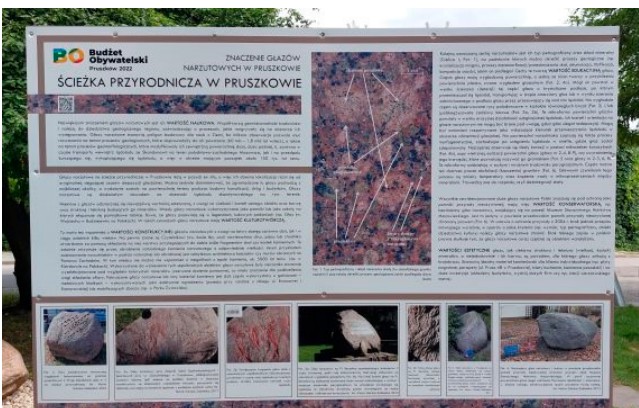

**Figure 11.** Information board entitled "Significance of Erratic Boulders in Pruszków".

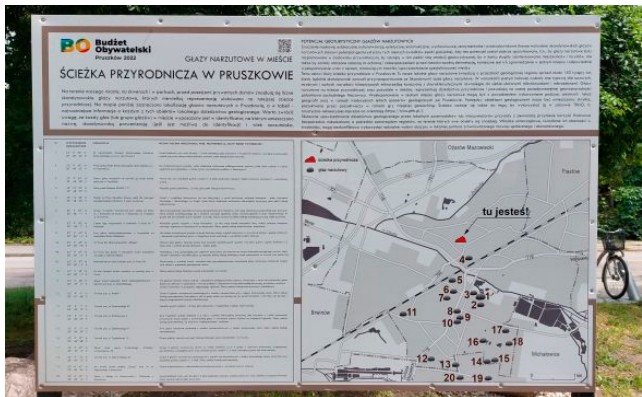

**Figure 12.** Information board entitled "Erratic Boulders in the City".

It is worth mentioning that on the basis of the lapidarium, geological knowledge (geo-interpretation) has already been transferred to the recipients several times. So far, only the authoress of this article has geo-interpreted the features on the Geosite. In the first year of the Geotrail's operation, she carried out several geo-interpretations of the geosite for schoolchildren (pro bono publico) (Figure 13). However, the project promoter aims to ensure that the geosite's high educational value is used by local geography teachers themselves (e.g., [11]). In order to achieve this, the authoress of the project has invited them to methodological workshops (Figure 14). Furthermore, the geosite is open to the general public during two or three "summer nature walks around the city", which the authoress has been organizing almost every year since 2012. Her promotion of the city's geoheritage draws the participation of approximately 10–20 individuals each time (Figure 15). There is no charge for visiting the geosite or taking part in the summer nature walks.

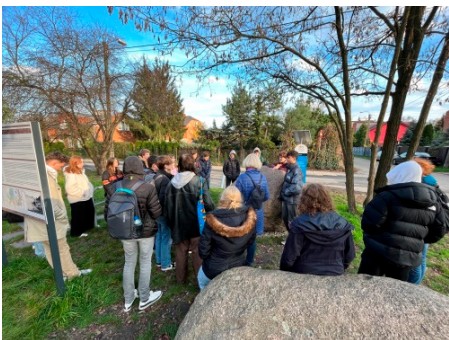

**Figure 13.** Geography lesson on the nature trail for students of a secondary school in Pruszków; photo M. Nowak 2022.

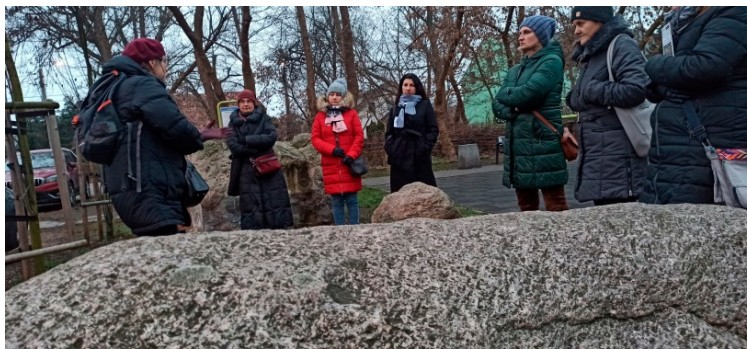

**Figure 14.** Educational workshops on the nature trail for geography teachers from the Pruszków district. Photo by R. Szewczyk, 2022.

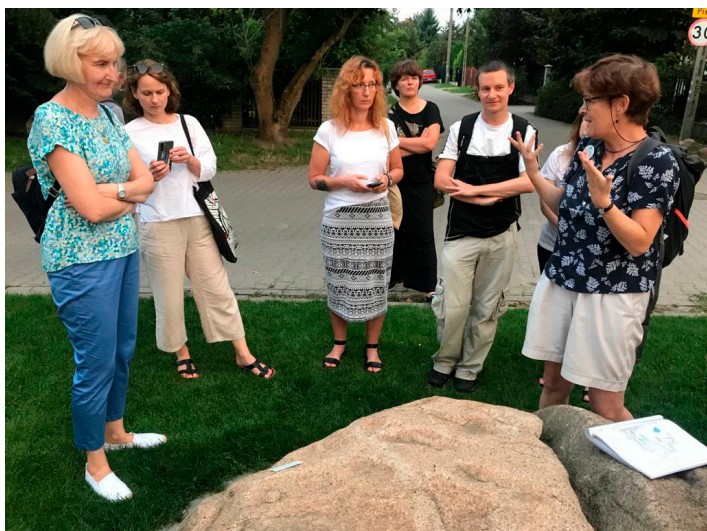

**Figure 15.** A summer nature walk in Pruszków (29 August 2023), during which the authoress discussed the geoheritage of Central Mazovia.

The abiotic trail is intended for all Pruszków residents who wish to study the geological history of the region and its Mazovian geoheritage, exhibit interest in Pruszków's geoheritage, value the conservation of non-living nature, and desire an exceptional outdoor experience to enrich their understanding of the history preserved in stone.

*4.3. Lapidarium in Reguły, Michałowice Commune (No. 3)*

The lapidarium in front of the Michałowice Municipal Office building in Reguły contains eight erratic boulders (Figure 16, Table 2): igneous (Nos. 1, 3–4, 6–8), and metamorphic (No. 2, 5).

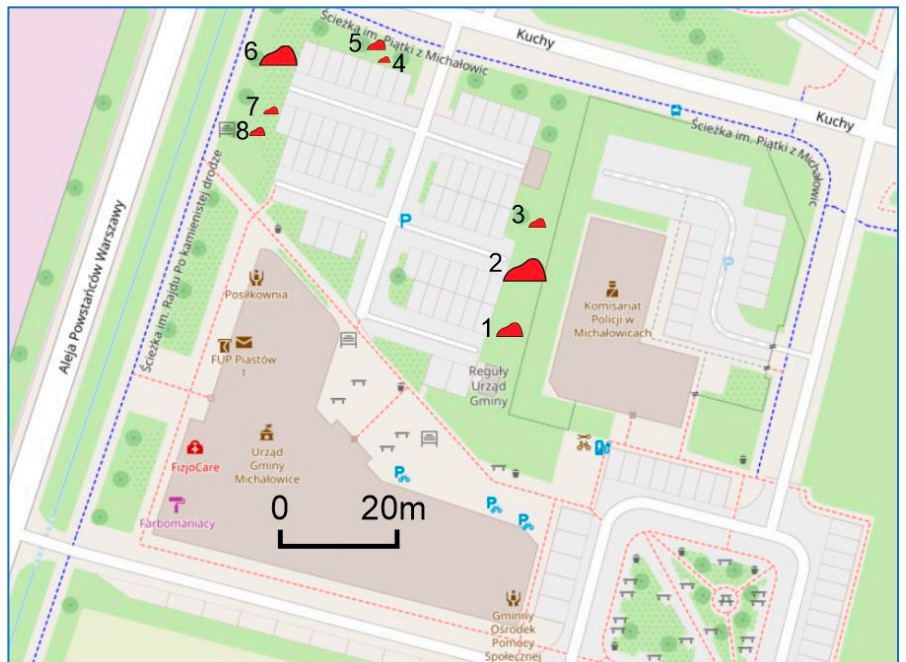

**Figure 16.** Plan of the lapidarium in front of the building of the Michałowice Commune office in Reguły. Relevant numbers are listed in Table 2.

**Table 2.** Information about erratic boulders in the lapidarium in front of the building of the Michałowice Municipal Office in Reguły.

| No. | Vol. [m³] | Weight [t] | Petrographic Type of the Rock, Name of the Indicator Erratic and Its Scandinavian Provenance, Characteristics of the Specimen's Surface, Age |
|---|---|---|---|
| 1 | 0.56 | 1.54 | Småland granite from south-eastern Sweden, weathered rock fracture surface on car park side, slight eolisation near ground surface, locally colonised by lichens; age: 1.45–1.35 B |
| 2 | 1.93 | 5.32 | Gneiss with slightly eolised surface, surface exfoliation in the upper part of the rock, locally colonised by lichens |
| 3 | 0.18 | 0.49 | Småland granite from southeastern Sweden, with abrasive edges, locally colonised by lichens; age: 1.45–1.35 B |
| 4 | 0.08 | 0.21 | Åland rapakivi granite from the Åland Islands in the central Baltic, abraded edges, on the upper surface legible glacial striae; age: 1.7–1.5 B |
| 5 | 0.27 | 0.75 | Granitogneiss with vein, upper surface formed by glacial polish with clear glacial striae |
| 6 | 0.89 | 2.45 | Granite with vein, upper surface formed by glacial polish with concentric, crescentic burrs |
| 7 | 0.09 | 0.24 | Coarse-grained granite with abraded edges |
| 8 | 0.11 | 0.30 | Åland quartz granite from the Åland Islands in the central Baltic; highly weathered with eolian eroded cavities of non-erosion-resistant feldspar; abraded edges; age: 1.7–1.5 B |

Inventory: Maria Górska-Zabielska 2022.

Among the erratic boulders collected, there are four indicator erratics, i.e., rocks for which we know the exact area of origin (Table 2, Figure 17). These include easily recognisable erratics from the Åland Islands: Åland rapakivi granite (No. 4; Figure 18) and Åland quartz granite (No. 8). This group also includes Småland granites (Nos. 1 and 3).

The collections of erratic boulders in front of the Michałowice Municipal Office come from the surrounding area and were brought to the surface during the construction of sewers, roads and buildings.

The boulders are easily accessible (also to people with disabilities), you can approach them directly and, thanks to the educational and information board nearby, you can identify the petrographic type and mineral composition (Figure 18), which can be used to determine the geological processes (e.g., magma crystallisation, metamorphic processes, accumulation, lithification and sediment compaction) to which they were subjected.

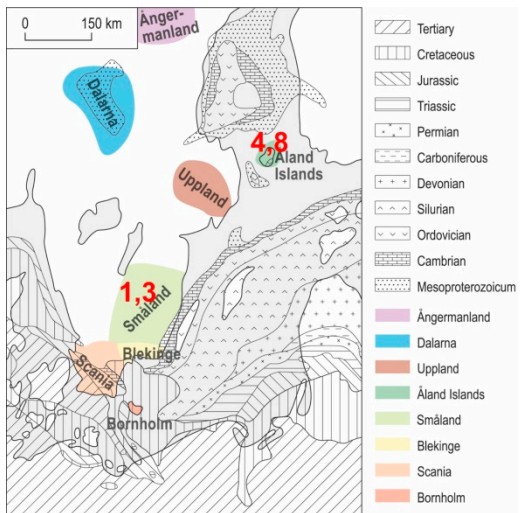

**Figure 17.** Source areas of the indicator erratics that can be found in the lapidarium situated in front of the building of the Michałowice Commune Office, serving as a main Scandinavian maintenance area. Relevant numerical data are consistent with Table 2.

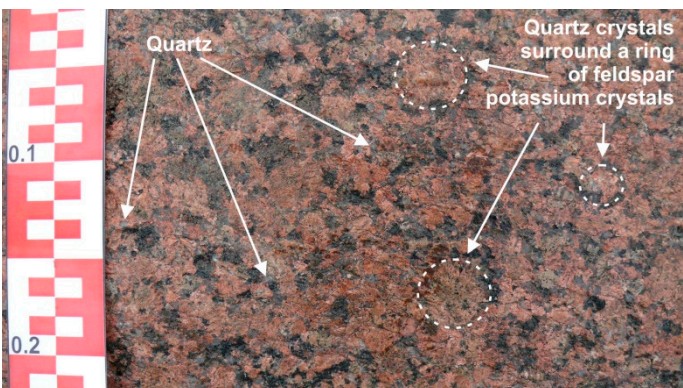

**Figure 18.** Rapakivi granite from the Åland Islands present in erratic boulder No. 4, located in front of the Michałowice Commune Office building. The rock's rapakivi structure is highlighted by dashed boundaries, and the entire surface of the stone exhibits evenly distributed smoky-gray quartz crystals (follow arrows).

The boulders often have a rounded silhouette and abraded edges (Nos. 3–8; Figures 19 and 20), typical features indicating glacial transport. Perfectly surrounded by the energetic water environment of the intra-glacial tunnels, the boulder takes on the shape of a rounded object (Nos. 4, 8).

An erratic boulder's side may develop a relatively smooth area, known as a glacial polish (Nos. 4–6; Figures 19 and 20). Such a feature could have resulted from the detersion of this part of the boulder against the crystalline substrate over which the ice sheet was transported the boulder, or from the abrasion of the boulder that was anchored in the substrate while the ice sheet moved over it. Postglacial characteristics, such as parallel furrows (numbers 4 and 5, depicted in Figure 19) and co-concentric, crescent-shaped markings (number 6, depicted in Figure 21), are frequently found on the rock surface. These microforms are caused by the subglacial ice sheet's erosion activity. Since the boulder has been relocated, its configuration and orientation cannot be considered. However, these observations can suggest the direction of the ice sheet's movement in the Scandinavian region.

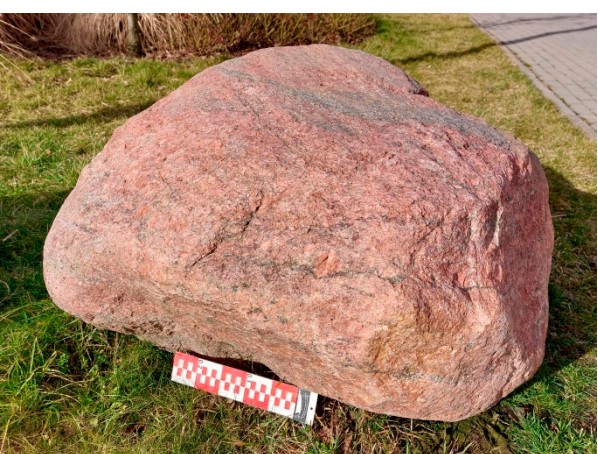

**Figure 19.** The upper surface of a granite gneiss (No. 5) exhibiting a glacial polish with distinct parallel scratches, formed during the transportation of the boulder on rough, hard terrain. Additionally, abraded edges suggest glacial transport.

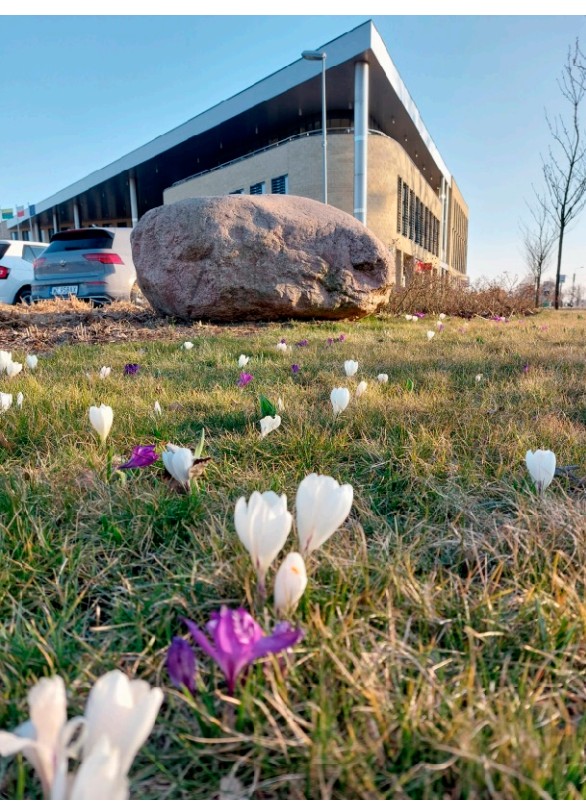

**Figure 20.** Erratic boulder No. 6, a granite rock that is well covered and has visible glacial polish from the pedestrian zone. On its surface, there are co-concentric and crescentic marks (Figure 21) that provide a record of the processes that occurred at the bottom of the ice sheet.

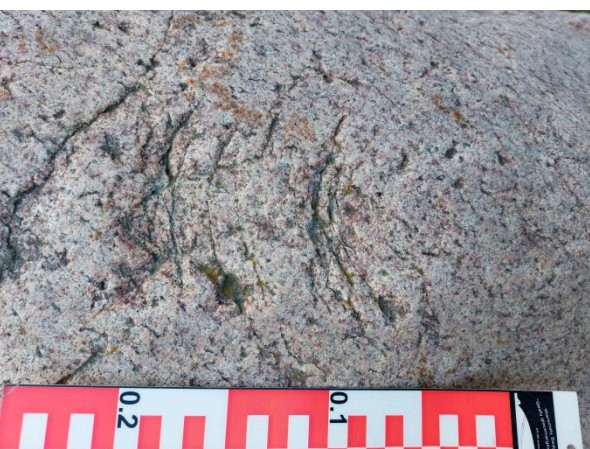

**Figure 21.** Concentric and crescent-shaped marks that appear on the surface of erratic boulder No. 6. These markings are believed to be the result of glacial erosion, as the boulder was transported by ice across the landscape.

The surface of the erratic boulder also documents morphogenetic processes that occurred following the withdrawal of the ice sheet in the area where the boulder was deposited. The most common traces of corrosion (the grinding of orographic obstacles by quartz grains moving in wind currents) are observed in the form of corrosion micro-reliefs (No. 8; Figure 22), pox and the aeolization of the boulder surface (Nos. 1 and 2). The effects of destructive processes in a dry and frosty periglacial environment are most effectively recorded in the form of a faceted crest. The rock is then called a faceted boulder (ventifact).

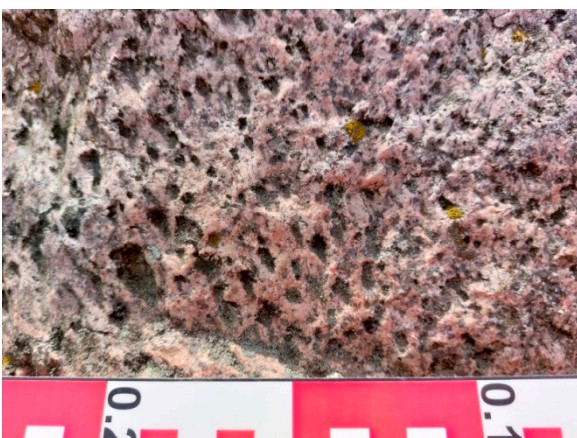

**Figure 22.** Cavities with a distinctly elongated shape, resulting from the loss of weathered minerals through corrosion, serving as evidence of periglacial activity in the foreland of the retreating ice sheet. This is exemplified by boulder No. 8.

The exfoliation (flaking) of rocks is a frequently observed process (No. 2; Figure 23). Temperature changes and water circulation in the microspaces among the minerals are the primary contributors to this phenomenon. These cause the rock to disintegrate.

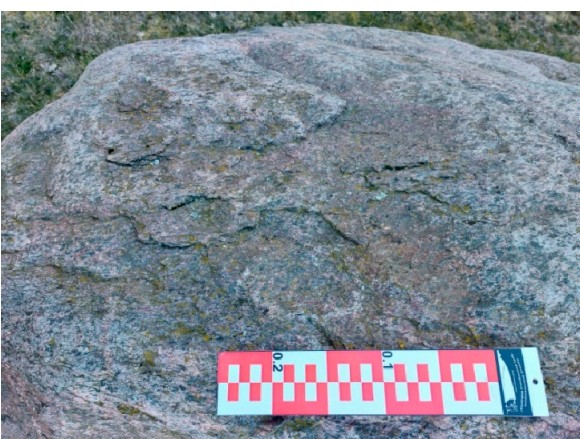

**Figure 23.** A record of exfoliation observed in the uppermost section of gneiss boulder No. 2, resulting in a progressively rounder appearance of the object.

It is also difficult to ignore the ecological function of collected boulders when they become a habitat for lithophyte (epilithic and endolithic) flora, which adhere to the substrate with their grackles (Nos. 1–3; Figure 24). The two elements of this relationship, the rock and the lichens/mosses, are linked by an antagonistic coexistence (parasitism), with one benefiting from direct contact and the other suffering a loss (e.g., the destruction of rocks). The type of rock has little influence on the species of plant that colonise it [42]. Lichen type and size are fairly good bioindicators of air pollution. As lichens obtain water and nutrients directly from dry and wet precipitation, air quality can be assessed. Lichen type and size should be compared with the lichen scale (see Attachment 3: lichen scale in [83]). The bushier the lichen, the more the air represents the normal vegetation zone.

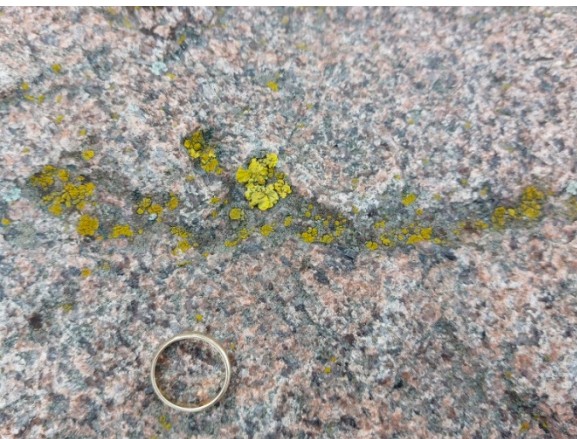

**Figure 24.** Epilithic vegetation, made up of *Pleurococcus* and *Xanthoria parietina*, which often colonises the erratic boulders and finds on them an ecological niche in which it can survive.

*4.4. Lapidarium at the Top of the Komorów River Dam Lake, Michałowice Commune (No. 4)*

A collection of erratic boulders placed on the crest of the Komorów river dam lake (on the Utrata River) in the Commune of Michałowice comprises sixteen erratic boulders (Figure 25). There are deep igneous rocks (Nos. 1–3, 5–9, and 12–16), which are light metamorphosed (Nos. 7 and 11), and metamorphic (No. 10). There is also one representative of clastic sedimentary rocks (No. 4), derived from the layers covering the Baltic Shield. More of the detailed information of the boulder collection is given in Table 3.

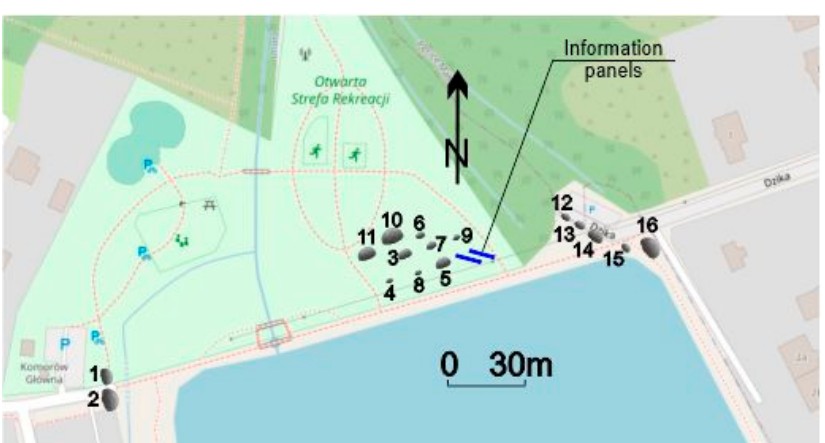

**Figure 25.** Plan of the lapidarium at the top of the Komorów river dam lake. Relevant numerical data are consistent with Table 3.

**Table 3.** Information about erratic boulders in the lapidarium at the top of the Komorów river dam lake.

| No. | Vol. [m³] | Weight [t] | Petrographic Type of the Rock, Name of the Indicator Erratic and Its Scandinavian Provenance, Characteristics of the Specimen's Surface, Age |
|---|---|---|---|
| 1 | 0.36 | 0.98 | Karlshamn granite from southern Sweden with large feldspars of tabular habit; age: 1.45–1.35 B |
| 2 | 0.2 | 2.24 | Småland granite from southeastern Sweden with an eolised surface; age: 1.75–1.5 B |
| 3 | 0.09 | 0.24 | Veined microgranite |
| 4 | 0.01 | 0.04 | Jotnian sandstone, eolian ventifact; age: 1.3 B; Figure 26 |
| 5 | 0.14 | 0.38 | Småland granite from southeastern Sweden with cavities from loss of weathered feldspar; age: 1.75–1.5 B |

**Table 3.** *Cont.*

| No. | Vol. [m³] | Weight [t] | Petrographic Type of the Rock, Name of the Indicator Erratic and Its Scandinavian Provenance, Characteristics of the Specimen's Surface, Age |
|---|---|---|---|
| 6 | 0.07 | 0.19 | Microgranite, very well abraded, exfoliated in the upper part of the rock; Figures 27 and 28 |
| 7 | 0.06 | 0.17 | Microgranite, gneissitic in places, with veins, glacial polish in upper part of rock |
| 8 | 0.05 | 0.13 | Småland granite from southeastern Sweden; age: 1.75–1.5 B |
| 9 | 0.07 | 0.19 | Coarse-grained granite with tabular feldspars, highly weathered with erosion-resistant feldspar crystals protruding above the rock surface |
| 10 | 0.63 | 1.72 | Gneiss with fold |
| 11 | 0.39 | 1.07 | Granitogneiss, strongly eolised, in the upper part of the rock exfoliated |
| 12 | 0.08 | 0.21 | Microgranite, with veins, partly abraded with a good edge abrasion, eolised |
| 13 | 0.29 | 0.81 | Åland quartz granite, eolised over large area, poorly exposed glacial polish from Komorów Lagoon; age: 1.7–1.5 B |
| 14 | 0.55 | 1.51 | Microgranite, exfoliated over large area, colonised by small epilithic flora |
| 15 | 0.22 | 0.61 | Åland quartz granite, very well abraded, eolised on the lagoon side, and here a visible corrasive microsculpture; age: 1.7–1.5 B; Figure 29 |
| 16 | 0.67 | 1.84 | Åland granite with a clear rapakivi structure; age: 1.7–1.5 B; Figure 18 |

Inventory: Maria Górska-Zabielska 2022.

The rock garden has been created according to the principles of the creation and management of walking paths [27,80,82]. It is located in the immediate vicinity of the outdoor fitness centre and is accessible to people with disabilities.

Some erratic boulders collected here have worn edges (Nos. 6 and 15, Table 3; Figures 27–29), a typical feature of glacial transport. Perfectly surrounded by the energetic water environment of the intra-glacial tunnels, the boulder assumes the shape of a rimmed object (Figure 27).

Glacial polish can be seen on one of the walls of erratic blocks (Nos. 7 and 13), which is a record of the processes that took place in the subglacial zone of the ice sheet.

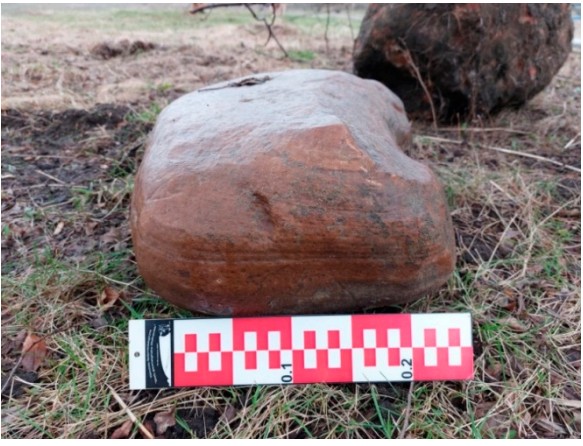

**Figure 26.** Iotnian Sandstone (No. 4) is the singular sedimentary rock example in the lapidarium. A corroded facete is present on the upper surface of the rock.

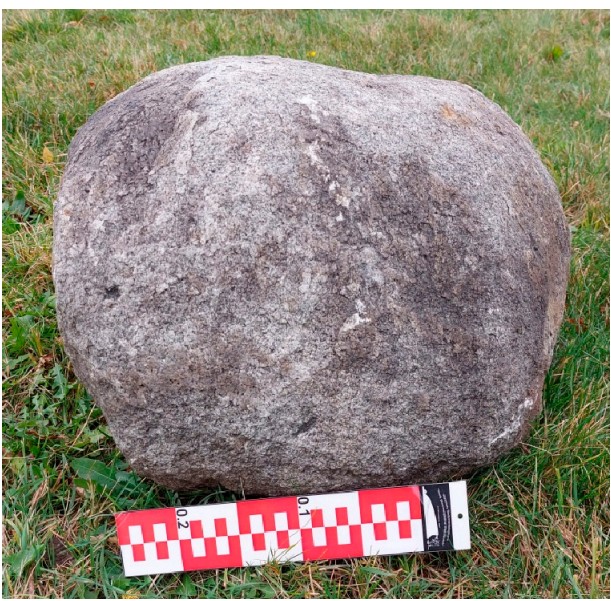

**Figure 27.** A microgranite pebble (No. 6), with a rounded shape attributed to the effect of the vigorous water surroundings in the intra-glacial tunnels.

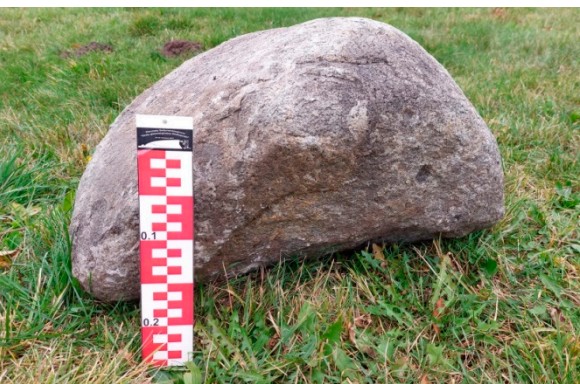

**Figure 28.** In the upper part of the microgranite (no. 6), traces of an exfoliation process can be observed. This process also leads to a gradual rounding of the boulder's shape.

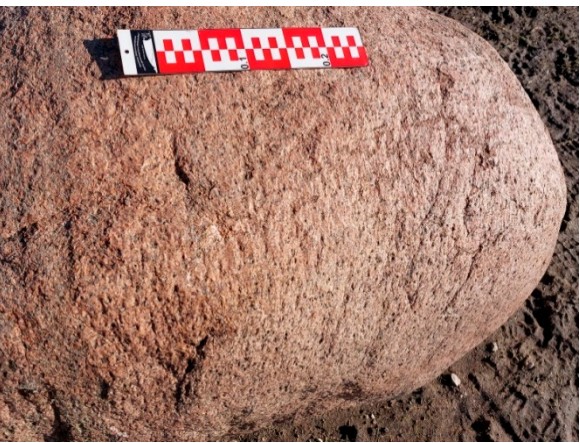

**Figure 29.** Micro-ridges and micro-corrosion grooves observed on erratic boulder No. 15, providing evidence of periglacial processes in the foreland of the retreating ice sheet.

Morphogenetic processes typical of the periglacial zone (the foreland of the retreating ice sheet) are also recorded on the surface of the ice sheets. The Komorów boulders show

traces of corrosion in the form of a micro-relief (No. 15; Figure 29), small stains, the aeolization of the boulder surface (Nos. 2, 11–13 and 15) and a polished facete (No. 4; Figure 26). In the last case, we are talking about a ventifact.

Exfoliation on the surface of the granites (Nos. 6, 11 and 14; Figures 27 and 28) is observable due to changes in temperature along with the circulation of water, mainly rain, in the micro-spaces amidst the minerals. Amongst the 16 boulders, 5 have a weight of more than a tonne (Nos. 2, 10, 11, 14 and 16)—their size, structure, texture, and often their colour cause them to disappear from the landscape, hence attracting interest. Nevertheless, they should serve the next generations of Poles. Unfortunately, this is the result of inadequate ecological education in schools and, as a consequence, inappropriate pro-environmental attitudes among adults. Will five boulders from Komorów, weighing more than 1 tonne, share this fate?

Erratic boulder No. 14 in the local collection has an ecological function because it is a habitat for epilithic flora that clings to its surface with its claws.

There are seven indicator erratics among the collected geo-items. These are Åland rapakivi granite (No. 16; Figure 30) and Åland quartz granite (Nos. 13 and 15). This group also includes Småland granite (Nos. 2, 5 and 8) and Karlshamn granite (No. 1). The sedimentary erratic—Iotnian sandstone (No. 4; Figures 26 and 30)—comes from the bottom of the Baltic Sea basin. We do not call this rock an indicator erratic because, due to methodological limitations, it is not possible to point to a specific place where it was ploughed.

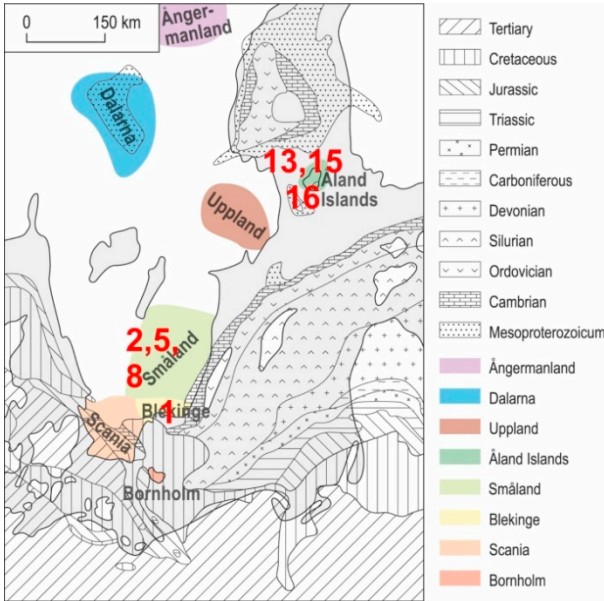

**Figure 30.** The sources of the indicator erratics in the lapidarium at the top of the Komorów river dam lake compared with those in other significant Scandinavian parent regions. The corresponding figures can be found in Table 3.

### 4.5. The "Jędrek" Boulder in Wilkowyja, Garwolin Commune (No. 5)

Until 2018, the "Jędrek" boulder was lying in situ on arable land, where it obstructed the field work of its owner, Andrzej Konopacki. Influenced by Henryk Szarek, a local resident, geography teacher and initiator of the project, who was aware of the role and importance of this geo-object, the owner decided to donate it to the local community free of charge. Together with the headmistress of the school at the time, Iwona Kurowska, they had the attention of the authoress of the article, who supported them meritoriously. Thanks to the financial support of the then mayor of the Commune, Marcin Kołodziejczyk, and the logistical support of the Wilkowyja Fire Brigade and local construction and transport companies, which lent their equipment, the boulder was dug out of the tilly sediments in

March 2018 and transported to its current location. It now stands in front of Maria Wójcik's Primary School in Wilkowyje. The boulder "Jędrek" (named after the donor—Jędrek is an abbreviation of the name Andrzej) was legally protected as a monument of inanimate nature by a resolution of the Garwolin Commune Council adopted on 8 June 2018 (Figure 31a,b; no entry in the GDOŚ register). Since that day, it has been under protection.

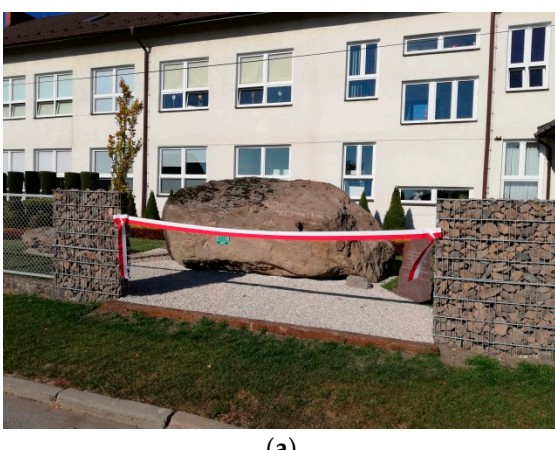 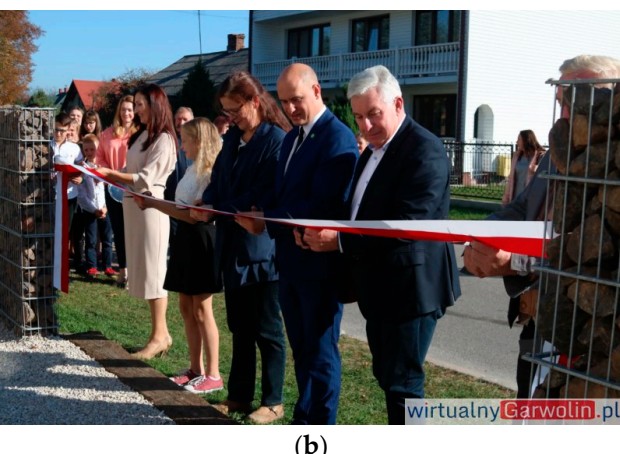

(**a**)   (**b**)

**Figure 31.** (**a**,**b**) Handing over ceremony of the erratic boulder "Jędrek" as a new monument of inanimate nature in Mazovia to the local community in Wilkowyja, 16 October 2018.

On the day of the ceremonial unveiling of "Jędrek", the authoress gave a geoeducational lecture to the assembled school community and the inhabitants of the village.

The dimensions of "Jędrek" are impressive and make up its exceptional value. It is a gneiss with a calculated volume of 13.6 m$^3$ and a weight of 37.39 tonnes. It is an integral aspect of the region's geological heritage and adds to its geodiversity. Together with the meandering Wilga River, which in places cuts into the gravel alluvium, formed dunes on the foreland of the retreating ice sheet, and the flat-bottomed ground moraine with glacial till, now transformed into brown soil, this boulder constitutes the foremost evidence of the geological phenomena that occurred in the Garwolin area in the past and, more broadly, in central Mazovia. Such properties are integral to the scientific value of the item. Initially, H. Szarek was looking for the educational function of "Jędrek"; he included it in the geological educational trail around Wilkowyja, along which the students of the school learn about the local geoheritage. Today, other teachers at the school follow in his footsteps.

Located in the immediate vicinity of the school, "Jędrek" has a guaranteed long-term educational role in developing appropriate pro-environmental attitudes among students. Located in a representative place in Wilkowyja, surrounded by gabions filled with much smaller erratics, it increases the aesthetic value of the surrounding area. "Jędrek" also has a cultural function—it was erected to commemorate the 100th anniversary of Poland's independence and the founding of the school in front of which it stands. An oak tree planted at that time and a commemorative plaque complete the exposition of the boulder. It has a sentimental function for the donor. The huge erratic boulder will remain in Wilkowyje's history for centuries.

## 5. Discussion

Novel and innovative methods of communicating Earth and environmental science knowledge by an experienced geo-interpreter outside of school/work, during free time from compulsory lessons and pleasantly spent in the field, perhaps interpolated as part of some other activity, have a high chance of reaching, inspiring interest in and engaging the recipient. Children, here, are a special group. Early childhood education in natural settings supports the long-term development of pro-environmental skills, competencies, and behaviours that endure through to adulthood and have a cascading effect on others [84].

In the case of five new facilities in a peripheral tourist area, such activities can initiate geotourism, a tool for the sustainable development of local government units and a driving force/lever of the local economy. Development opportunities are most often seen in the development of tourism, which will provide a source of income for local residents. However, in the case of peripheral or indifferent tourist areas, geotourism comes to the rescue, when it is based on the geoeducation and geoprotection of the objects present in the surroundings that testify to the geological and geomorphological past of the region. There is evidence in the literature that geotourism can significantly improve the quality and comfort of life of those involved in the presentation of geo-objects and, consequently, as geointerpreters disseminate their presence and indicate the role they play in the proper functioning of the local environment (e.g., [25,33,85–87]). It should also be remembered that geotourism is characterised by 1. environmental responsibility—it is committed to protecting resources and preserving geodiversity; 2. cultural responsibility—it is committed to respecting local sensitivities and benefiting from local geodiversity.

The paper discusses five new geoheritage objects in Central Mazovia. These are two huge erratic boulders in situ in Żochy and Wilkowyja, and three lapidaries in Pruszków, Michałowice and Komorów, which serve as educational trails. All five provide an in situ interpretation and ex situ museum display of geology [44]. Irrespective of their exposure, they all play a scientific and cognitive role—they provide an opportunity to satisfy human needs for learning about the inanimate natural heritage of the region. Rock gardens serve multiple purposes through their exhibits. The rocks gathered there signify the geological inheritance of the region and add to its geodiversity. They record various geological processes that once occurred in the area from which they originated and present a geological exhibition for the region. It is not only the proximity of schools that ensures that the facilities fulfil an educational function. Geoeducation can be a form of lifelong learning because it is not limited to the walls of the classroom. Geocommunication is closely linked to the development of appropriate pro-environmental attitudes and includes aspects of geoconservation. The collections collect and protect several erratic rocks from the surrounding area from destruction and environmental degradation.

These interesting erratic boulder installations and the knowledge gained about them build a territorial identity with the region and fill local communities with pride. This translates into an openness and lack of resistance from local communities to further developments of this kind. The best evidence of this is the requests made to the authoress to support the creation of new lapidaries and geological trails. The authoress is currently co-ordinating work on four new sites in different parts of Poland.

There is some concern that a geological trail located in an uninteresting part of the city without supervision (e.g., video surveillance) could be destroyed. Of the five geosites, only at the Pruszków lapidarium were two information boards destroyed. Thanks to the support of the municipality, the boards were replaced with new ones. The authoress hopes that they will now be able to provide long-term and unhindered use to be enjoyed by local residents, schoolchildren and tourists.

Geo-objects placed in harmony with the surroundings increase the aesthetic value of the area, square or pocket garden (e.g., [68,88–90]). At the same time, they provide a psychological benefit to people who are in an emotionally disturbed state. They can be a destination for recreational activities such as walking or cycling.

According to [91], there is an indisputable need to strengthen geoenvironmental education in the curriculum. It is crucial for understanding the Earth's complex processes, natural resources and their interactions with human activities. Such training provides people with the knowledge and sensitivity necessary to understand the environmental problems facing us and the measures needed to live together in a sustainable manner. By incorporating geoenvironmental education into the curriculum, we enable future generations to make informed decisions, act responsibly and help protect our planet for both present and future generations. This need is becoming increasingly apparent as global environmental problems continue to worsen, underscoring the urgent need for comprehensive education

that bridges the gap between geological understanding and sustainable environmental management.

## 6. Conclusions

The chapter concentrates on the key findings. It focuses on geoeducation and sustainable geotourism and gives practical conclusions.

1. The geo-entities mentioned in this paper, located in Middle Mazovia in central Poland, have the potential to serve as comprehensive tools with which to reach out to the general public and bridge the gap between laypeople and the field of geoscience. The chief aims of the regional geoheritage showcased in the paper and promulgated in the local community include the following:

   - Creating a place for education, recreation and relaxation;
   - Getting to know the geological heritage of the region in which they live;
   - Raising awareness of the unique geological past of Central Mazovia;
   - Creating a territorial identity within the region;
   - Drawing attention to the problem of protecting inanimate nature;
   - Sustainable development, which has the potential to shape attitudes and behaviour towards the geo-environment and cultivate geo-ethical values (cf. [5,10,14]).

2. All these activities are part of the development strategies of Mazovian local authorities. Properly exposed natural sites that are inanimate play a crucial role in preserving and enhancing the geographical character of a place. This includes its environment, culture, aesthetics, heritage and the wellbeing of its inhabitants (cf. [41]).

3. Four out of five of the geological sites are shaping the right pro-environmental, ecological attitude of the local inhabitants and tourists, who want to learn about the geological past of the region and the local geodiversity of Mazovia, who are sensitive to the protection of inanimate nature and who want to spend time outdoors in a unique way and deepen their knowledge of the history written in stone. The collections of erratic boulders were created in accordance with the principles of creating and managing hiking trails (e.g., [27,80–82,92]). The facilities (with the exception of No. 1) are accessible to people with reduced mobility.

4. The described geosites are located in a peripheral tourist area, which, in the light of research (e.g., [46–50]), does not prevent the inhabitants from aspiring to live in smart cities. They satisfy the increasingly sophisticated needs of society through tourism, wellbeing and geo-ecosystem benefits. Finally, the smart management of natural resources, through citizen participation and/or expert assistance, can be a driver for innovation, with the overall goal of sustainable economic development and high quality of life (e.g., [25,33,86,87]).

5. It cannot be overlooked that local initiatives raising awareness among the population and promoting all the values of geotourism will certainly help to raise awareness of the need to protect the Earth's inanimate resources more than has been the case to date. Nevertheless, the authoress believes that each human being cares about a clean, beautiful and recognised little homeland.

**Funding:** This research was funded by Jan Kochanowski University in Kielce, Poland, grant number SUPB.RN.21.256. The authoress 's project, the Pruszków nature trail, was established using funds from the city's 2022 civic budget. Both lapidaries in Michałowice Commune were created thanks to subsidies from the Michałowice Municipality Office.

**Data Availability Statement:** All data collected, as part of the field and in-camera work, is provided in this text.

**Acknowledgments:** My gratitude is owed to my husband, Ryszard Zabielski of the Polish Geological Institute—National Research Institute, for his invaluable assistance during my fieldwork.

**Conflicts of Interest:** The authoress declares no conflict of interest.

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
