# Peer review of "New Geoeducational Facilities in Central Mazovia (Poland) Disseminate Knowledge about Local Geoheritage"

_sustainability, doi:10.3390/su152216115_

Round 1

Reviewer 1 Report

Comments and Suggestions for Authors

The manuscript underscores the role of newly unveiled geoeducational facilities in Central Mazovia, elucidating local geoheritage and bolstering earth science education and regional economies. However, some suggestions could help to improve the clarity of the research methodology and refine the conclusions:

Methodology Enhancements:

Structured Presentation: Outline data collection and analysis processes clearly, ensuring a comprehensive and replicable methodology.

Techniques Clarification: Specify the tools, software, or techniques utilized and explain their application in detail.

Geo-Object Categorization: Explain the classification criteria and discuss the integration of secondary data sources, if applicable.

Conclusion Refinements:

Focus and Brevity: Streamline the conclusion to focus on key findings, ensuring conciseness while retaining critical information.

Quotation Integration: Consider paraphrasing the included quotation for seamless integration and flow.

Actionable Insights: Highlight the study's practical implications and actionable insights, focusing on geoeducation and geotourism impacts.

 The manuscript's value in geoeducation will surge with refined methodology and concise conclusions for enhanced reader engagement and academic impact.

Author Response

Dear Reviewer, 
Please read the attached response to your comments and suggestions, 
Yours sincerely

Reviewer 2 Report

Comments and Suggestions for Authors

The article is quite well written and addresses a contemporary issue.

The author(s) must illustrate implications of the study explicitly.

How the study outcomes can contribute to the body of knowledge?

Comments on the Quality of English Language

N/A

Author Response

(The authors gave the same response as above.)

Reviewer 3 Report

Comments and Suggestions for Authors

The article concentrates on the underdeveloped school education in geological heritage and geodiversity which certainly also has influences on adult life. It emphasizes the necessity of an attractive geo-education that has to be offered inside and outside of schools. A particularly important role in this respect  is played by geo-interpreters or geo-tour guides. The author concentrates in her geological research on the area of Central Mazovia in Poland and is in her presentation able to deal with five new geo-objects that "have a great potential for geo-education, for spreading knowledge about local geodiversity and for attracting the interest of geotourists and the general  public." These five locations in Central Mazovia, in three towns and two villages, and open to the public,  offer erratic boulders with considerable geo-educational value, which are described in detail. The article shows quite well in which way geological information can be processed and made available for a broader public and, in particular, for school education. The article would, in my opinion, still be more valuable for a larger and international readership, if it will deal a bit  more with the way and occurring support and/or  problems and resistance in the creation of the geo-educational sites and their long-term usage.

Comments on the Quality of English Language

For me, as not an English  native speaker, there are no real problems. But still, a copy editor should check the paper.

Author Response

(The authors gave the same response as above.)

Reviewer 4 Report

Comments and Suggestions for Authors

Dear Author,

I was asked to read and revise your article “New geoeducational facilities in Central Mazovia disseminate knowledge about local geoheritage”, you proposed for publication to Sustainability.

The paper described five geosites, their scientific, didactic and touristic relevance, and the way they were made exploitable to the local communities.

In my opinion, the data presented and discussed in the article are relevant and exemplificative case studies, which can be considered for publication, although some minor adjustments to the MS are needed, thus I am proposing the Editor to accept it after minor revisions.

For my line-by-line comments, please refer to the attached file.

I am not a native English, but I found the text plain and fluent. Some minor editing changes are needed, for example regarding the figures and their captions.

The MS does not follow the “classical” codified organization adopted in scientific papers, but this choice is justified by the very nature of the topic dealt with, for which to keep separated “Materials and Methods” and “Results” is difficult. Nonetheless, some parts could be moved, and the final section could be better named “Discussion and conclusions”.

I also noticed some inaccuracies in the use of geological terms (a sandstone is a clastic sedimentary rock, pebble refers to a clast size regardless of its shape, some of the objects figured are probably larger than bolders, a gneiss is a medium to high degree metamorphic rock that could be hardly considered not fully metamorphic), as well as some misleading sentences (e.g., I agree erratics in Northern Europe come from the Scandinavian area, but they are related in a general way to glacial processes and they are not specific to a geographical area).

Anyway, I found the article really relevant within topics such as geoheritage, geoeducation and involvement of the local population in the knowledge and conservation of geological assets.

I fully agree with many of the reflections you proposed, as they are more or less applicable to many different local contexts and regardless of country.

The involvement of the citizens, starting from school age, in the valorization of the territory they live in and what it has to offer to the quality of their lives is crucial for the success of any long-lasting (geo)conservation project.

Hope my comments could have been useful at this stage.

Kind Regards

The reviewer

Author Response

Dear Reviewer,

I appreciate all your comments/remarks/corrections in the *.pdf file very, very much. I have followed them and have corrected my text.

However, I also have some remarks according to:

 - Fig. 1,

 - lines 329-341,

 - line 367, first comment,

 - lines 607-609.

I have commented on your suggestions in the *.pdf file. I have attached it for your convenience,

Bows,
the authoress
